# Current Understanding of and Future Directions for Endometriosis-Related Infertility Research with a Focus on Ferroptosis

**DOI:** 10.3390/diagnostics13111926

**Published:** 2023-05-31

**Authors:** Hiroshi Kobayashi, Chiharu Yoshimoto, Sho Matsubara, Hiroshi Shigetomi, Shogo Imanaka

**Affiliations:** 1Department of Gynecology and Reproductive Medicine, Ms.Clinic MayOne, 871-1 Shijo-cho, Kashihara 634-0813, Japan; shogo_0723@naramed-u.ac.jp; 2Department of Obstetrics and Gynecology, Nara Medical University, 840 Shijo-cho, Kashihara 634-8522, Japan; chiharu-y@naramed-u.ac.jp (C.Y.); s.matsubara@kei-oushin.jp (S.M.); hshige35@gmail.com (H.S.); 3Department of Obstetrics and Gynecology, Nara Prefecture General Medical Center, 2-897-5 Shichijyonishi-machi, Nara 630-8581, Japan; 4Department of Medicine, Kei Oushin Clinic, 5-2-6 Naruo-cho, Nishinomiya 663-8184, Japan; 5Department of Gynecology and Reproductive Medicine, Aska Ladies Clinic, 3-3-17 Kitatomigaoka-cho, Nara 634-0001, Japan

**Keywords:** assisted reproductive technology, endometriosis, ferroptosis, granulosa cells, infertility

## Abstract

Background: To date, the development of therapy for endometriosis and disease-related infertility remains a major challenge. Iron overload caused by periodic bleeding is a hallmark of endometriosis. Ferroptosis is an iron- and lipid-reactive oxygen species-dependent type of programmed cell death that is distinct from apoptosis, necrosis, and autophagy. This review summarizes the current understanding of and future directions for the research and treatment of endometriosis and disease-related infertility, with the main focus on the molecular basis of ferroptosis in endometriotic and granulosa cells. Methods: Papers published between 2000 and 2022 in the PubMed and Google Scholar databases were included in this review. Results: Emerging evidence suggests that ferroptosis is closely linked to the pathophysiology of endometriosis. Endometriotic cells are characterized by ferroptosis resistance, whereas granulosa cells remain highly susceptible to ferroptosis, suggesting that the regulation of ferroptosis is utilized as an interventional target for research into the treatment of endometriosis and disease-related infertility. New therapeutic strategies are urgently needed to efficiently kill endometriotic cells while protecting granulosa cells. Conclusions: An analysis of the ferroptosis pathway in in vitro, in vivo, and animal research enhances our understanding of the pathogenesis of this disease. Here, we discuss the role of ferroptosis modulators as a research approach and potential novel treatment for endometriosis and disease-related infertility.

## 1. Introduction

Endometriosis is an estrogen-dependent disease characterized by the presence of endometrial-like glands and stroma outside the uterus [1]. It affects approximately 10% of women of reproductive age. The underlying etiology of endometriosis is heterogeneous and diverse, with three phenotypes: superficial peritoneal disease, ovarian endometrioma, and deep infiltrating endometriosis [2]. Endometriosis can cause pelvic pain, dysmenorrhea, dyspareunia, and infertility [1]. Tanbo and Fedorcsak reported that endometriosis can cause pelvic adhesions, pain, and infertility by inducing acute and chronic inflammation [3]. Furthermore, Guo SW stated that endometriosis might be better redefined as “a condition that started with the ectopic deposition of endometrial stroma and epithelium which undergo cyclic bleeding and thus repeated tissue injury and repair, resulting in gradual and progressive smooth muscle metaplasia and fibrogenesis” [4], i.e., endometriosis-associated infertility and fibrosis may be two sides of the same coin. Indeed, 30–50% of women with endometriosis can be diagnosed with infertility, and endometriosis occurs in 25–50% of women with infertility [5]. Hormonal suppression or surgical lesion removal effectively relieves pain; however, hormonal treatment is unsuitable for women desiring pregnancy [6]. Endometriosis-related infertility is caused by reproductive tract anatomy (e.g., anatomic distortion and tubal occlusion due to pelvic adhesions), impaired steroidogenesis and fertilization, impaired follicular development due to folliculogenesis alteration, ovulation dysfunction, impaired oocyte and embryo quality, and impaired endometrial receptivity (embryo implantation defect) [3]. Assisted reproductive technology (ART) is considered an integral element of treatment for women with endometriosis-related infertility due to defects in oocyte maturation and low fertilization rates [7].

Sampson’s theory of retrograde menstruation and implantation is the most widely accepted mechanism of endometriosis development [2]. Erythrocytes and apoptotic endometrial tissues that transplant into the peritoneal cavity via retrograde menstruation are well-known inducers of inflammation, immune dysregulation, and oxidative stress, i.e., an oxidative/antioxidant imbalance [8,9]. Endometriosis has been associated with iron overload in the peritoneal cavity and endometriotic cysts [10,11]. Iron-induced oxidative stress has been found to play a key role in the pathogenesis of endometriosis and its related infertility in a mouse model [12]. Although the exact mechanism of human endometriosis-related infertility is unknown, recent studies suggest that ferroptosis closely contributes to the pathogenesis of endometriosis [11,13,14]. Ferroptosis is a type of iron-dependent programmed cell death that leads to the lethal accumulation of lipid peroxides. It closely contributes to the pathogenesis of many diseases, such as cancer, neurodegenerative diseases such as Parkinson’s and Alzheimer’s diseases, stroke, ischemia-reperfusion injury or ischemic organ damage, acute renal failure, hepatotoxicity, cardiovascular disease, and pulmonary fibrosis [15,16]. Several clinical trials focusing on ferroptosis are currently underway [17].

This review summarizes our current understanding of research in endometriosis and disease-related infertility, highlights the molecular mechanisms and signaling pathways involved in ferroptosis, and finally discusses future perspectives for therapeutic strategies.

## 2. Mechanisms Underlying Endometriosis-Related Infertility

### 2.1. Morphological Changes

This subsection summarizes morphological changes in the reproductive tract (anatomic distortion and tubal occlusion due to pelvic adhesions) and oocytes. Advanced endometriosis sometimes affects the morphology of pelvic tissues, including occluded fallopian tubes, due to adhesions [3]. The mechanical disruption causes reduced ability for oocyte release and embryo implantation, therefore leading to reduced reproductive performance [3]. A significant increase in oocyte follicle atresia in the affected ovary was observed [18]. Histopathological and biological assessment of oocytes affected by endometriosis provides important information on morphological patterns, including zona pellucida hardening, nuclear anomaly, cytoplasmic granularity, spindle disruption, and mitochondrial abnormalities [19,20]. Women with endometriosis had significantly increased incidence of impaired cumulus-oocyte complex (COC) expansion and decreased first polar body extrusion [21]. The affected oocytes exhibited increased zona pellucida hardening, thus preventing fertilization [22]. Cell nuclei were characterized by the presence of decentralized chromatin and abundant nucleoli [23]. The oocyte exhibited dysmorphisms, including cytoplasmic granularity and the presence of vacuoles [19]. It is also suggested that spindle disassembly or disruption does exist in in vitro maturation oocytes [22,23]. Furthermore, oocytes were likely to have lower cytoplasmic mitochondrial content, to have a lower number of mitochondrial DNA (mtDNA) copies, and to fail in in vitro maturation [20,23,24,25,26]. In summary, oocytes affected by endometriosis may exhibit morphological and functional defects.

### 2.2. Poor Oocyte Quality in Endometriosis

This subsection summarizes that impaired COC expansion, oocyte immaturity, and a reduction in the number of mature oocytes retrieved and fertilization rates are important causes of endometriosis-related infertility. A meta-analysis and prospective case–control studies demonstrated a reduction in the number of oocytes retrieved per cycle and in the percentage of mature oocytes in in vitro fertilization and embryo transfer (IVF-ET) patients with endometriosis compared with those with other causes of infertility [9,20,27,28,29,30]. Compared with the healthy ovary, the number of oocytes retrieved was significantly lower not only in the ovarian follicles in close proximity to the endometrioma but also in the distal follicles of the same ovary [28,29]. Furthermore, the number of oocytes retrieved was lower in the affected ovary than in the contralateral ovary [31]. In addition, the number of oocytes was lower in the ovary after laparoscopic excision of the endometrioma than in the contralateral normal ovary, possibly due to previous surgical treatments [32]. The reproductive outcome following IVF-ET was poorer in women who underwent surgical treatment for endometriosis than in those without endometriosis [30]. The number of oocytes retrieved was lower in women with endometriosis than in those without; thus, the maximum number of oocytes retrieved is the potential clinical marker for determining oocyte quality [33]. Therefore, oocyte immaturity is believed to be an important cause of endometriosis-related infertility.

In addition, a systematic review and meta-analysis of case–control studies revealed that women with endometriosis have lower fertilization rates and poorer IVF outcomes than those with other causes of infertility [34,35]. A large retrospective study based on the register from the American Society for Reproductive Medicine yielded similar results [36]. However, a systematic review and meta-analysis revealed that the reproductive outcomes (e.g., live birth rate) were similar between women with endometrioma who underwent IVF/intracytoplasmic sperm injection (ICSI) and those without endometrioma [37]. A case–control study demonstrated that the fertilization rate was significantly lower in the IVF-ET group (*p* < 0.03) than in the ICSI group (*p* = 0.38) [35]. In general, surgical treatment can lead to a diminished ovarian reserve [38]. Some studies showed that the fertilization rate was reduced in women who underwent surgery [39,40]; however, others showed no significant reduction [37,41]. These results indicated inconsistent reproductive outcomes in terms of whether the fertilization rate was reduced in women who underwent surgical treatment for moderate and severe endometriosis. There were also interesting studies that reported that the fertilization rate was lower in women with minimal and mild endometriosis than in those with moderate and severe endometriosis [20,37]. On the other hand, Yang et al. summarized that the implantation and live birth rates were similar in women with and without endometriosis [29].

Interestingly, women who received oocytes from women who were endometriosis donors had significantly lower implantation and pregnancy rates than those who received oocytes from women who were non-endometriosis donors; however, no significant difference was observed in the implantation and pregnancy rates between endometriosis recipients and non-endometriosis recipients [42,43]. The qualities of oocytes and embryos as well as uterine receptivity influence the success of IVF-ET treatment, whereas endometriosis mainly adversely affects oocyte quality. Taken together, the infertility associated with endometriosis may be caused by poor oocyte quality.

## 3. Oxidative–Antioxidant (Redox) Balance in Endometriosis

Oxidative stress is defined as an imbalance between reactive oxygen species (ROS) production and the antioxidant defense system. ROS have both beneficial (e.g., a modulator of cell proliferation) and deleterious (e.g., an inducer of lipid, protein, and DNA damage) effects. ROS are inflammatory mediators that fine-tune cell survival and death. Furthermore, ROS regulate a variety of physiologic functions, e.g., oocyte maturation, steroidogenesis, ovulation, and corpus luteal function [8]. However, excessive oxidative stress leads to decreased oocyte quality [22]. Therefore, ROS may be implicated in the pathophysiology of endometriosis [8,43,44,45] and infertility associated with this disease [22,46].

### 3.1. Biological Markers Involved in Redox Balance in Endometriosis

This subsection summarizes the biomarkers evaluated in in vitro, in vivo, and human studies on oxidative stress in the serum, follicular fluid, granulosa cells, and oocytes of affected women (Figure 1). Serum triglycerides, total cholesterol, and low-density lipoprotein levels increased and high-density lipoprotein levels decreased in women with endometriosis [47]. Lipid hydroperoxides (LOOHs) are formed by the endogenous oxidative degradation of membrane lipids via lipid peroxidation (LPO) [47]. The level of malondialdehyde (MDA), an index of LPO, is higher in the serum of women with endometriosis than that of controls, indicating the presence of oxidative stress [48]. Follicular fluid composed of granulosa cell secretions and plasma exudates is essential for the creation of an intrafollicular microenvironment that controls oocyte growth and development. Many researchers observed a significant increase in ROS, nitric oxide (NO), MDA, and 8-hydroxy-2′-deoxyguanosine (8-OHdG), an indicator of oxidative DNA damage, in the follicular fluid of women with endometriosis [44,49,50,51]. The presence of oxidative DNA damage may be associated with the pathogenesis of endometriosis-related infertility possibly through the promotion of chromosomal instability, meiotic abnormalities, spindle disruption, and LPO and therefore reduction of oocyte quality [44,50,52,53,54]. Excessive oxidative stress-induced LPO may cause cell damage and death. Thus, follicular fluid oxidative stress may be useful as a diagnostic marker for oocyte quality prediction. However, inconsistency still exists, as Nakagawa et al. demonstrated that the oxidative stress status was not increased in the follicular fluid of patients with endometriosis [55]. Furthermore, intracellular ROS levels are found to be elevated in the granulosa cells of women with endometriosis [27]. Additionally, ROS and oxidative dysregulation of NO are responsible for poor oocyte quality by increasing protein nitration [22].

### 3.2. Oxidative Stress Caused by Iron Overload

We investigated the mechanism by which many oxidative stress markers are elevated in endometriosis. This subsection summarizes oxidative stress caused by iron overload in endometriosis. Iron is an essential element for the biological processes (e.g., oxygen transport and storage in tissues, hormone synthesis, energy metabolism in the mitochondria, ATP generation, mediation of oxidation–reduction reactions, nitrogen fixation, and DNA synthesis and repair) of almost all living cells and organisms [56]. Contrary to these beneficial features, iron overload amplifies intracellular oxidative stress via the Fenton reaction (Fe^2+^ + H_2_O_2_ → Fe^3+^ + OH^−^ + OH); causes damage to DNA, lipids, and proteins; and exerts cytotoxic effects on living cells [10]. Retrograde menstruation, the process in which menstrual blood including erythrocytes, macrophages, and endometrial tissue are transported through the fallopian tubes to the peritoneal cavity, is the key feature in understanding the pathogenesis of endometriosis [8,9]. Furthermore, periodic hemorrhage from ectopic endometriotic lesions is a hallmark of endometriosis and causes iron overload [9]. Erythrocyte-derived iron is one of the well-known inducers of oxidative stress [8,9]. Altered iron metabolism in response to iron overload may be associated with the development and progression of endometriosis [57]. At moderate levels, iron-induced ROS activate ectopic endometrial cell proliferation, angiogenesis, and adhesion. Indeed, the animal model showed that the number and size of endometriotic lesions in the iron-treated mice were significantly greater than those in the control mice [58]. Several lines of evidence support that an imbalance in iron homeostasis may regulate endometriotic cell proliferation [57,58].

Indeed, studies on the expression of iron and hemoglobin species demonstrated the presence of hemoglobin, heme, or free iron in the endometriotic ovarian cyst, peritoneal fluid, or follicular fluid of patients with endometriosis [10,27,57,59,60,61] (Figure 1). The concentration of free iron in endometriotic cysts was 100 to 1000 times higher than that in peripheral blood or in other types of benign cysts [28,62,63]. The iron content in follicles adjacent to the endometriotic cysts was higher than that in contralateral intact ovaries [28]. Total iron content was ranked from highest to lowest as follows: endometrioma proximal follicles, endometrioma distal follicles, and healthy ovarian follicles [28]. These data suggest that iron contained in ovarian endometriomas can freely diffuse out from the cyst wall and can reach the granulosa cells [54,64]. However, one study showed no difference in the iron content of the follicular fluid between the affected and contralateral ovaries [64]. In addition to iron, follicular fluid H and L ferritin contents, transferrin saturation, and transferrin receptor 1 (TFR1) expression were higher in the affected ovaries than in the contralateral ovaries [28]. Transferrin, which is mainly produced in the liver and extra-hepatically, acts as an iron carrier and may suppress ROS generation [65]. TFR1 expressed on the surface of granulosa cells mediates cellular iron uptake via endocytosis of iron-loaded transferrin [65]. The concentration of follicular fluid transferrin decreased in women with endometriosis-related infertility [61]. Excess iron may reduce transferrin concentration in follicular fluid via increased transferrin saturation. Iron overload and transferrin insufficiency induce the excess of ROS, compromising the integrity of the mitotic spindle by promoting chromosome instability, which may affect the number and maturation of oocytes retrieved from women with endometriosis [27,61]. On the other hand, endometriotic macrophages act as iron scavengers by up-regulating a variety of scavenger receptors, such as transferrin receptor expression [26]. Macrophages contain hemosiderin but also produce heme and iron that cause deleterious ROS [8,26]. Collectively, the high iron content of ovarian endometriomas adversely affects the adjacent granulosa cells through oxidative stress by free iron-mediated Fenton reactions, reducing the quantity and quality of the retrieved oocytes [28,54,63], which could be associated with impaired fertility and adverse pregnancy outcomes [23,66].

### 3.3. Dysregulated Antioxidant Systems

We then summarize the antioxidant factors that influence the determination of the redox balance in endometriosis. First, glutathione (GSH) is a master antioxidant that protects cells against oxidative damage. Glutathione peroxidase (GPx) is a GSH-dependent antioxidant enzyme. Some proteins of the GPx family are differentially expressed in the follicular fluid during the menstrual cycle in women of reproductive age [67]. The GPx concentration in rat ovaries and human follicular fluid was the highest during ovulation [67]. The levels of GSH and proteins of the GPx family were positively correlated with the oocyte maturation rate or the number of high-quality embryos [68]. Furthermore, paraoxonase 1 (PON1) is an antioxidant enzyme that degrades lipid peroxides in lipoproteins and cells [47]. Similar to GPx, the serum concentrations of PON1 increased at ovulation [47]. These results indicate that enzymatic antioxidants, such as GPx and PON1, play an important role in protecting cells from oxidative stress in human ovulation.

Second, cellular redox status was evaluated in both endometrial ectopic and eutopic tissues in patients with endometriosis compared with the control endometrium of healthy women [9]. ROS include superoxide anion (O_2_^−^), hydrogen peroxide (H_2_O_2_), and hydroxyl radical [10]. Superoxide anions are increased in both eutopic and ectopic endometrial cells than in controls, whereas H_2_O_2_ is higher in endometriotic cells than in controls and eutopic endometrial cells [69]. In addition, GPx, superoxide dismutase (SOD), and catalase play important roles in the endogenous antioxidant defense system against cellular oxidative stress. Research showed that the expression of GPx during ovulation and in the early secretory phase is lower in eutopic endometrial cells than in controls [70] (Figure 1). It has been reported that SOD activity is higher in ectopic endometria than in eutopic endometria [71]. Furthermore, Catalase expression is lower in endometriotic cells than in controls [69].

Third, we summarize the balance of oxidative–antioxidant systems in the follicular fluid of women with endometriosis. The follicular fluid of women with endometriosis contains elevated levels of pro-oxidants (e.g., ROS, NO, LPO, and iron) and down-regulated antioxidants (e.g., glutathione, GPx, SOD, catalase, and glutathione reductase) [9,49,68]. The concentrations of nonenzymatic antioxidants (e.g., vitamins A, C, and E) are lower in the follicular fluid of women with endometriosis than in controls [71,72]. The balance between pro-oxidants and antioxidants in endometriosis-related infertility was shown to be shifted toward oxidative stress in comparison with matched tubal infertility controls.

Finally, several candidate genes and proteins produced by granulosa or cumulus cells that activate oocyte protective pathways to prevent oxidative damage may improve the success rate of IVF-ET [73]. Molecules identified as candidate genes include antioxidants (e.g., glutathione, SOD, and aldehyde dehydrogenase (ALDH)) [74]. SOD1 is highly expressed in the cumulus cells of women with infertility and moderate and severe endometriosis and is positively associated with oocyte quality [48] or clinical pregnancy rate following ICSI [75]. ALDH (e.g., ALDH3, member A2 [ALDH3A2]) detoxifies lipid aldehydes [76]. The synthesis of ALDH3A2 in granulosa cells decreases with age and in women with endometriosis [76]. Decreased antioxidant production in granulosa cells could result in an imbalance of ROS with antioxidant cellular defenses, leading to endometriosis-related infertility via a state of excess oxidative stress [77].

In summary, each antioxidant has an emerging unique role in the pathogenesis of endometriosis. Follicular fluid adjacent to endometriosis is an oxidative environment that may cause defects in the antioxidant system of granulosa and cumulus cells, disrupting the intrafollicular environment.

## 4. Ferroptosis

### 4.1. The Ferroptosis-Related Pathway

Endometriosis is characterized by iron overload caused by ectopic periodic bleeding. Excess iron catalyzes the Fenton reaction to generate free radicals and then oxidative stress-induced LPO. Oxidative stress is the major cause of cell death, including apoptosis, autophagy, and ferroptosis, with the latter being an iron- and ROS-dependent form [16]. Therefore, endometriosis is thought to be closely associated with ferroptosis mediated by iron-dependent oxidative stress [16,78,79]. Accumulating evidence suggests that ferroptosis is implicated in pathological conditions, such as cancer, neurodegenerative diseases, ischemic organ injuries, and pulmonary fibrosis [11,13,27,78,80]. Regarding these studies, we summarize the distinctive molecular mechanisms of several key pathways in the regulation of ferroptosis in endometriotic and granulosa cells as well as endometriosis-associated macrophages. Molecules that play key roles in ferroptosis are involved in metabolic and biochemical processes, including iron homeostasis, LPO, glutathione metabolism, cysteine exploitation, mevalonate cholesterol biosynthesis, and nicotinamide adenine dinucleotide phosphate (NADPH) function [81].

Figure 2 illustrates the ferroptosis-related pathways in granulosa cells, of which eight seem to play a major role in orchestrating ferroptosis. High levels of ferric iron (Fe^3+^) present in the extracellular space (e.g., follicular fluid) bind to transferrin, promoting transferrin saturation and internalization into the cytoplasm via the TFR (Figure 2, ①). Ferric iron is converted to ferrous iron (Fe^2+^), and ferrous iron accumulates in the cytosolic iron pool as ferritin, an intracellular iron storage protein. Furthermore, ferritin increases the iron concentration in the cytoplasm by mediating iron-selective autophagy (i.e., autophagic degradation of ferritin, ferritinophagy) [82] (Figure 2, ②). Ferritinophagy plays a central role in the control of granulosa cell ferroptosis [27]. Granulosa cell ferritinophagy is a key factor in promoting ferroptosis. Genes related to ferritinophagy include up-regulated autophagy-related gene 5 (ATG5), nuclear receptor coactivator 4 (NCOA4), and ferritin heavy chain 1 (FTH1) expression as well as down-regulated GPx4 and tumor suppressor protein p53 expression [27]. NCOA4 facilitates ferritin binding and mediates ferritinophagy, releasing labile-free iron via lysosomal degradation of ferritin. These data suggest that iron overload in follicular fluid and granulosa cells promotes NCOA4 expression in granulosa cells, leading to active ferritinophagy and enhancing LPO via the Fenton reaction. Ferroportin, a known vertebrate iron exporter, exports iron from iron-storing cells, such as hepatocytes, enterocytes, macrophages, endometriotic cells, and granulosa cells [83] (Figure 2, ③). Contrarily, hepcidin blocks the export of cellular iron by inducing the internalization and degradation of ferroportin [83]. Excess iron potentiates the generation of free radicals via the Fenton reaction, damaging lipids, proteins, and DNA (Figure 2, ④). Furthermore, iron induces LPO of polyunsaturated fatty acids (PUFAs), eventually leading to ferroptosis [14]. On the other hand, the GSH-dependent GPx4, as an antioxidant system, is important for the regulation of ROS and prevents the buildup of lipid peroxides by reducing H_2_O_2_ (Figure 2, ⑤). Cystine, an essential precursor for GSH biosynthesis, is transported into cells via the cystine/glutamate antiporter SLC7A11 (also known as xCT) (Figure 2, ⑥). Glutathione and GPx4 exert protective effects against oxidative stress [84]. Isopentenyl pyrophosphate (IPP), a key product of the mevalonate cholesterol biosynthesis pathway, is essential for GPx4 synthesis [85] (Figure 2, ⑦). Therefore, mevalonate pathway inhibition reduces GPx4 synthesis and promotes ferroptosis-dependent injury [85]. NADPH is essential for counteracting oxidative stress and maintaining cellular redox homeostasis [84] (Figure 2, ⑧). NADPH depletion leads to a decrease in intracellular glutathione and impairs redox homeostasis [84].

### 4.2. The Ferroptosis Pathway in Endometriosis

This section summarizes how ferroptosis contributes to the pathophysiology of endometriosis, granulosa cells adjacent to endometriosis, and endometriosis-associated macrophages.

Figure 3 presents a comparison of ferroptosis-related pathways in endometriotic and granulosa cells. The left image illustrates how endometrial cells can survive under oxidative stress and harsh environments, focusing on ferroptosis. Gene expression profile datasets showed that the expression of ferroptosis-related genes in patients with endometriosis increased in the order of normal endometrium, eutopic endometrium, and ectopic endometrium [80], suggesting that resistance to ferroptosis is a hallmark of endometriosis. Indeed, ferroptosis induced by iron overload was confirmed in endometriotic stromal cells in contact with endometrioma cyst fluid [78]. However, endometriotic cells acquire mechanisms for resistance to ferroptosis via up-regulation of GPx4 and its upstream regulatory target GSH [70]. Recently, Fibulin 1 (FBLN1) has been discovered as a key ferroptosis-resistant candidate molecule in endometriosis [86]. It has been identified as a tumor suppressor in several cancers, e.g., gastric, prostate, breast, and ovarian cancers [87]. FBLN1 promotes cell proliferation and migration in endometriotic stromal cells by inhibiting ferroptosis [86]. Furthermore, the activation of the cholesterol mevalonate biosynthetic pathway may suppress ferroptosis via GPx4 activation, suggesting a close relationship between the mevalonate pathway and ferroptosis signaling in endometriotic cells [79,88]. Resistance to ferroptosis is commonly enhanced in endometriotic cells, allowing them to survive, progress, and establish endometriotic lesions [9,79]. Interestingly, a study also demonstrated that ferroptosis in endometriotic cells may promote the angiogenesis of adjacent lesions by triggering cytokine secretion as a paracrine action [78]. Thus, ferroptosis may cause endometriotic cell death but also contributes to the progression of endometriosis. The mechanisms of ferroptosis and ferroptosis resistance in endometriosis are still poorly understood.

### 4.3. The Ferroptosis Pathway in Granulosa Cells

The right image in Figure 3 illustrates the ferroptosis pathway in granulosa cells. Granulosa cells or surrounding cumulus cells in follicles play a key role in oocyte maturation, which relies on the intrafollicular environment. Excess iron and transferrin deficiency were both identified in the follicular fluid of women with both infertility and endometriosis [61]. Generally, granulosa cell ferroptosis causes an imbalance between the concentrations of iron and transferrin, impairing oocyte quality and adversely affecting reproduction [27]. The granulosa cells of women with endometriosis had increased MDA, a metabolite of LPO products, but decreased regulatory inhibitors of the ferroptosis activation pathway (e.g., glutathione and GPx4) compared with controls. Ferroportin exports iron from granulosa cells, and hepcidin blocks cellular iron release through ferroportin [83]. In the granulosa cells of women with infertility, ferroportin showed decreased mRNA levels [89]. Contrarily, serum hepcidin levels were found to be increased in women with infertility [89]. Increased hepcidin and ferroportin deficiency may cause iron overload, accelerating ferroptosis in granulosa cells. Ferroptosis is characterized by functional (e.g., increased LPO) and morphological changes in mitochondria (e.g., fragmentation and cristae enlargement) and thus may also play a critical role in the pathogenesis of endometriosis-related infertility [11,27,90,91]. In addition, iron overload in peritoneal fluid and endometriotic cysts decreases GPx4 expression and induces LPO [11]. These data suggest that iron overload in follicular fluid and granulosa cells can induce ferroptosis, impairing oocyte maturation, disrupting blastocyst formation, and consequently leading to infertility [11]. Therefore, endometriotic cells acquire resistance to ferroptosis to avoid cell death, whereas granulosa cells are sensitive to ferroptosis.

### 4.4. The Ferroptosis Pathway in Macrophages

Macrophages contribute to the development of endometriosis by promoting angiogenesis. A recent study demonstrated that iron-dependent ferroptosis can up-regulate interleukin-8 and vascular endothelial growth factor-A expression in endometrial stromal cells and the macrophage THP-1 cell line [92]. Furthermore, iron overload suppresses the phagocytosis of macrophages [93]. Macrophages with decreased phagocytic activity are unable to eliminate ectopic endometrial cells. Iron-dependent ferroptosis may promote the development of endometriosis by impairing macrophage phagocytosis and producing more angiogenic cytokines [92].

Taken together, analysis of ferroptosis pathways revealed that differential susceptibility to ferroptotic responses exists between endometriotic and granulosa cells. Endometrial cells rely on a GPx-dependent antioxidant system to acquire intrinsic resistance to ferroptosis. On the other hand, granulosa cells often fail to sufficiently inhibit lipid ROS accumulation through the down-regulation of GPx expression and are prone to ferroptosis. These results indicate that endometriotic and granulosa cells are resistant and sensitive to ferroptosis, respectively (Figure 3). Enhanced susceptibility to ferroptosis in granulosa cells may lead to endometriosis-related infertility.

## 5. Therapeutic Strategies Targeting Ferroptosis

Sensitivity toward ferroptosis is regulated by metabolic pathways associated with iron, GSH/GPx4, mevalonate, PUFAs, and NADPH. Therapeutic strategies targeting ferroptosis hold great promise in the preclinical studies of various pathological processes, such as cancer, neurodegenerative diseases, acute renal failure, hepatotoxicity, and cardiovascular disease [14]. Extensive research has found that ferroptosis provides new opportunities for cancer therapy [94]. At least two types of ferroptosis modulators have been developed: ferroptosis inducers or activators as novel anticancer agents and ferroptosis inhibitors as novel cell protective agents. For example, cancer cells have been able to adapt to high ROS levels by activating their antioxidant systems [95]; therefore, ferroptosis inducers may be effective in cancer therapy by triggering excessive oxidative stress. Erastin, a ferroptosis activator, promotes lipid ROS accumulation by lowering intracellular glutathione and GPx4 via down-regulation of the cystine/glutamate transport receptor (system xCT) expression [96]. For example, elastin has been suggested to be a promising target for cancer therapy owing to its potent anticancer effects in breast cancer cell lines [97].

Furthermore, in vitro and animal studies have highlighted ferroptosis as a new therapeutic target for endometriosis [11,13]. GPx4, a key antioxidant defense enzyme that acts as a ferroptosis regulatory inhibitor, and its upstream regulatory target GSH were shown to be up-regulated in endometriotic cells [68] (Figure 3, left). Indeed, elastin-induced ferroptosis in ectopic endometrial stromal cells is demonstrated by elevated levels of iron, ROS, and lipid ROS [11,13]. Elastin also reduced the size of endometriotic lesions in mouse models [11,13]. Furthermore, simvastatin, a mevalonate pathway inhibitor, down-regulated the GPx4 expression, therefore inducing cancer cell ferroptosis and inhibiting endometriotic cell growth in the primate model [81]. Exogenous ferroptosis inducers are potentially good candidates that can accelerate lipid ROS accumulation and promote endometriotic cell death. On the other hand, unlike endometriotic cells, granulosa cells are characterized by reduced endogenous GPx4 expression [11]; therefore, ferroptosis inducers may promote further cell damage, leading to poor oocyte quality (Figure 3, right). Contrarily, ferroptosis inactivation was found to dramatically inhibit neuronal cell death, suggesting the beneficial and protective effects of ferroptosis inhibitors against neurodegenerative diseases [15]. Indeed, ferrostatin-1, a ferroptosis inhibitor, has been found to protect granulosa cells against oxidative damage caused by GPx4 deficiency [98]. Furthermore, Ni et al. demonstrated in mouse granulosa models that iron chelators and vitamin E are useful therapies for the treatment of endometriosis-related infertility [27]. However, there is currently no reliable evidence that ferroptosis inhibitors are useful for the aforementioned treatment. In addition, the long-term effects of ferroptosis inducers and inhibitors on endometriotic and granulosa cells as well as oocytes remain unknown, and many practical issues need to be resolved. Targeted therapies (inducers or inhibitors) against ferroptosis have only just begun in experimental endometriosis models, and there are currently no reports of clinical trials in patients with both infertility and endometriosis. 

## 6. Discussion

This review summarizes our current understanding of the mechanisms underlying endometriosis-related infertility and discusses potential future research directions and therapeutic strategies for ferroptosis modulators.

First, endometriosis causes a series of structural and functional changes to the reproductive system. Iron overload caused by periodic hemorrhage from ectopic lesions is a hallmark of endometriosis [9]. Bleeding may lead to the formation of adhesion, resulting in fibrosis. Infertility is common in women with endometriosis, and many patients benefit from ART, demonstrating a clear association between endometriosis and infertility. However, endometriosis has a detrimental impact (e.g., oxidative stress) on the local intrafollicular environment and adversely affects the clinical outcome of in vitro fertilization [54]. Several lines of evidence suggest that infertility associated with endometriosis, even minimal-to-mild disease, may be related to decreased oocyte quality [20,38,42,43]. The oxidative–antioxidant imbalance in endometriosis may result in substantial changes, including genetic alterations, in granulosa cells and adversely affect oocyte maturity [38,63,73]. The success of ART is influenced by gamete and embryo quality, but there are no reliable in vivo biomarkers that accurately reflect changes in quality [73]. The proteins in follicular fluid play essential roles in the regulation of follicular growth, follicular maturation, and ovulation by regulating inflammation, coagulation, complement, lipid metabolism, and antioxidant systems [67]. As follicular fluid is closely associated with granulosa cells and oocytes, iron- or oxidative stress-induced proteins (e.g., MDA, 8-OHdG, and ferroportin) may serve as biomarkers for evaluating oocyte quality [73]. Accumulating evidence suggests that the 8-OHdG level in follicular fluid is elevated in women with endometriosis [50] or negatively influences ICSI outcome [99]. The use of such markers in the follicular fluid may afford opportunities to predict and assess infertility outcomes.

Second, recent studies have focused on iron overload-dependent ferroptosis in the pathophysiology of endometriosis [70,78,80,86]. Selective induction of ferroptosis has emerged as a new treatment strategy in the mouse model of endometriosis [14]. Endometriosis is characterized by iron-dependent lipid peroxide accumulation and ferroptosis resistance through the up-regulation of redox enzymes, GPx4, and SOD [70]. In endometriotic cells, ferroptosis resistance plays a beneficial role in the maintenance of pathological functions in response to oxidative damage. The use of ferroptosis inducers may lead to the success of endometriotic cell death via lipid ROS accumulation. Contrarily, under iron-rich environments, granulosa cells also accumulate LPO but down-regulate ferroportin and GPx4 expressions, making them more susceptible to ferroptosis [11,59]. Overall, ferroptosis inducers are beneficial for the treatment of endometriosis in mouse models and in vitro studies but may be toxic to granulosa cells. The influence of ferroptosis inducers on endometriotic cell death and the molecular mechanisms behind how these drugs influence granulosa cells still warrant further studies.

Third, a ferroptosis inhibitor (e.g., ferrostatin-1) may be beneficial for the improvement of oocyte quality due to reduced oxidative damage in granulosa cells [98]. In preclinical studies, ferroptosis inhibitors have been recognized as useful drugs for the amelioration and treatment of various neurodegenerative diseases [100]. Hambright et al. reported that treatment with ferroptosis inhibitors improved neurodegeneration in mouse models of Alzheimer’s disease [100]. Furthermore, Santanam et al. showed that oral supplementation with vitamins C and E led to a significant decrease in the follicular fluid level of myeloperoxidase, a key mediator of oxidative stress response, in patients with severe endometriosis undergoing IVF [101]. Oral antioxidant therapy (e.g., oral melatonin supplementation) may be effective in reducing oxidative stress and improving oocyte quality [98]. Therefore, as future therapeutic strategies for infertility management, inhibitors of ferroptosis-related pathways may improve embryo quality and fertility outcomes by protecting granulosa cells and oocytes. However, whether ferroptosis inhibitor therapy exerts a beneficial effect on the fertility outcomes of women with endometriosis is still far from being proven.

Finally, we discuss the challenges facing new ferroptosis therapies. Many ferroptosis-related pathways, including TfR, ferroportin, xCT, GSH/GPx, and mevalonate, have been investigated as potential therapeutic targets for various diseases, including cancer [81]. The rationale for targeting the ferroptosis pathways in endometriosis is based on the preclinical data indicating that endometriosis is characterized by ferroptosis resistance [70,86]. On the other hand, granulosa cells adjacent to endometriosis are highly susceptible to ferroptosis [11,89]. Ferroptosis-induced granulosa cell damage or death consequently leads to certain pathological conditions, such as impaired oocyte quality and infertility. Aside from LPO suppression by ferroptosis inhibitors, several therapeutic strategies (ferritinophagy modulators, iron chelators, hepcidin inhibitors, modulators of ferroportin internalization, antioxidants [e.g., melatonin], and vitamin C and E supplementation) may positively affect the activation of granulosa cells and oocyte maturity (Figure 3). Ferroptosis inhibitors have been reported to be able to protect granulosa cells against oxidative damage [98]. However, endometriotic and granulosa cells must be strictly targeted by these ferroptosis modulators, as ferroptosis inducers or inhibitors positively or negatively alter lipid ROS balance. Ferroptosis modulators that target endometriotic lesions without harming granulosa cells are mandatory for future therapeutic strategies. Furthermore, the safety, tolerability, and efficacy of treatment with ferroptosis modulators in humans remain unknown.

In conclusion, the molecular mechanisms and signaling pathways involved in ferroptosis may be closely related to endometriosis and disease-related infertility. Furthermore, therapies targeting the ferroptosis pathway may hold promise in the future. Endometriosis is characterized by iron overload, excess oxidative stress, and ferroptosis resistance. Granulosa cells are highly susceptible to iron- and lipid ROS-dependent ferroptosis. An understanding of the latest progress in ferroptosis provides new opportunities for endometriosis and disease-related infertility research.

## Figures and Tables

**Figure 1 diagnostics-13-01926-f001:**
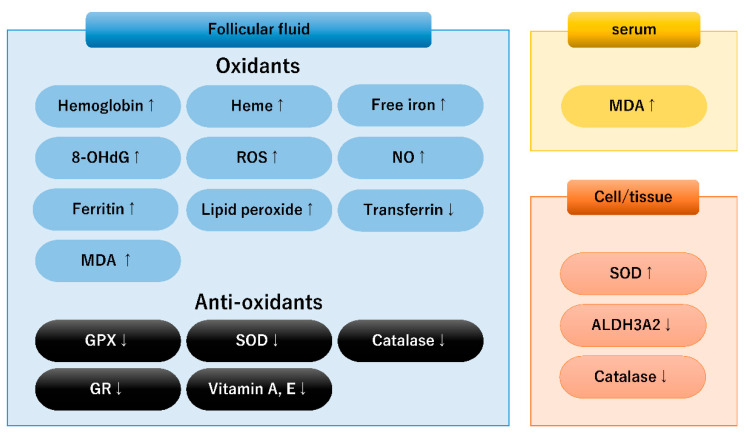
Candidate redox biomarkers in follicular fluid, serum, and granulosa cells of women with endometriosis. Up and down arrows indicate upregulation and downregulation, respectively. 8-OHdG, 8-hydroxy-2′-deoxyguanosine; ALDH3A2, aldehyde dehydrogenase 3, member A2; GPX, glutathione peroxidase; ROS, reactive oxygen species; GR, glutathione reductase; MDA, malondialdehyde; NO, nitric oxide; and SOD, superoxide dismutase.

**Figure 2 diagnostics-13-01926-f002:**
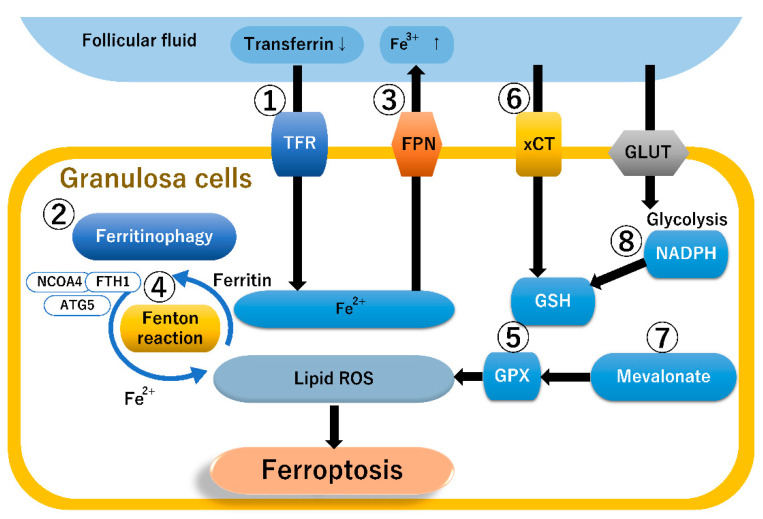
The ferroptosis-related pathways in granulosa cells. Up and down arrows indicate upregulation and downregulation, respectively. Please refer to the text for explanations of ① to ⑧ in the figure. ATG5, autophagy-related 5; FTH1, ferritin heavy chain 1; FPN, ferroportin; GLUT, glucose transporter; GPX, glutathione peroxidase; NADPH, nicotinamide adenine dinucleotide phosphate; NCOA4, nuclear receptor coactivator 4; TFR, transferrin receptor; and xCT, cystine/glutamate antiporter SLC7A11.

**Figure 3 diagnostics-13-01926-f003:**
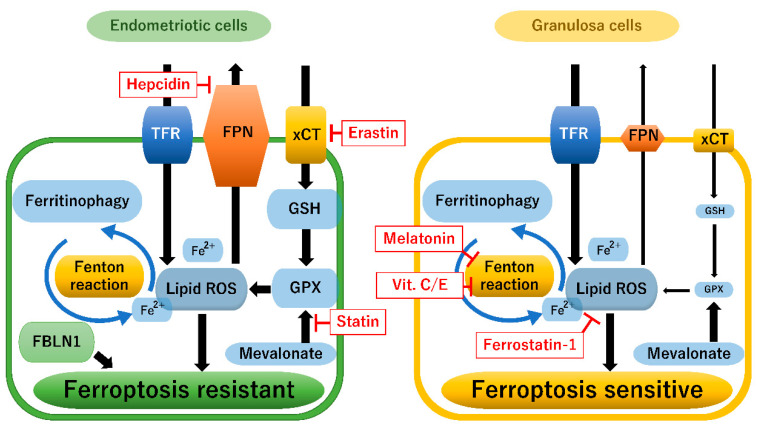
A comparison of the ferroptosis-related pathways in endometriotic cells and granulosa cells. Thin arrows indicate decreased activity compared to thick arrows. Boxed red letters indicate ferroptosis modulators. FBLN1, fibulin 1; FPN, ferroportin; GPX, glutathione peroxidase; GSH, glutathione; TFR, transferrin receptor; and xCT, cystine/glutamate antiporter SLC7A11.

## Data Availability

No new data were created.

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
