# Peer review of "Current Understanding of and Future Directions for Endometriosis-Related Infertility Research with a Focus on Ferroptosis"

_diagnostics, 2023, doi:10.3390/diagnostics13111926_

Round 1
Reviewer 1 Report
1. The authors present an overview of ferroptosis, an iron overload- and lipid-reactive oxygen species-dependent form of programmed cell death that is distinct from apoptosis, necrosis, and autophagy. The comments on treatment that suggest human data are overstated or under-referenced and need to be deleted or corrected.
2. Lines 27-28: “indicating that the regulation of ferroptosis can be utilized as an interventional target for treatment…” suggests there is human data. Consider “suggesting that the regulation of ferroptosis should be studied as an interventional target for treatment…” If you have human data, please clarify that.
3. Line 38: “The presence of endometrial glands and stroma” suggests that endometriosis is endometrium. “Endometrial-like,” “endometriotic,” and “similar to endometrial” suggest that endometriosis is not endometrium but may come from the endometrium, Müllerian rests, tubal metaplasia, peritoneum, bone-marrow stem cells or other. Consider “endometrial-like,” “endometriotic,” or “similar to endometrial” glands and stroma.
4. Line 42: “Endometriosis causes pelvic pain, dysmenorrhea, dyspareunia, and infertility.” Endometriosis does not cause all of those in all women with endometriosis. It can coincidental in fertile women at tubal ligation. Consider “can cause” in place of “causes.”
5. Line 47-48: Reference 5 (Tanbo et al.)is not correct for “Retrograde menstruation and periodic hemorrhage from ectopic lesions is a hallmark feature of endometriosis, leading to adhesion and fibrosis of surrounding tissues.” Tanbo et al. have “Once established, these hormone-responsive and cyclically active endometriotic lesions drive acute then chronic inflammatory reactions, and lead to pelvic adhesions, pain, and infertility.” See comments on repeated tissue injury and repair (ReTIAR) below. Please correct the reference or the statement.
6. Line 48, 493: Why don’t all women have adhesions if bleeding causes adhesions?
7. Lines 225-228: Sun-Wei Guo would add that the blood may come from wounds undergoing repeated tissue injury and repair (ReTIAR).
Guo SW, Ding D, Shen M, Liu X. Dating endometriotic ovarian cysts based on the content of cyst fluid and its potential clinical implications. Reprod Sci. 2015 Jul;22(7):873-83. doi: 10.1177/1933719115570907. Epub 2015 Feb 11. PMID: 25676579; PMCID: PMC4565481.
Guo SW. Fibrogenesis resulting from cyclic bleeding: The Holy Grail of the natural history of ectopic endometrium. Hum Reprod. 2018, 33(3):353-356. doi: 10.1093/humrep/dey015. PMID: 29420711
8. Lines 483-484: Change “infertility” to “mouse granulosa” in “Ni et al. demonstrated in infertility models…”
9. Lines 512-513: Do you mean “ferroptosis as the pathophysiology of endometriosis” or “ferroptosis in the pathophysiology of endometriosis”?
10. Lines 513-514: Ref 13. Liang et al., is a mouse model. Change “The selective induction of ferroptosis has emerged as a new treatment strategy in endometriosis [13]” to reflect the mouse model.
11. Line 519-520: There is no mention of treatment in reference 69, Ota et al. (2000). Reference 85, Wan et al. 2022, is not a treatment study. Delete or correct “Therefore, the activation of the ferroptosis pathways can be utilized as an interventional target for treatment of endometriosis.”
12. Line 523-524: “Overall, ferroptosis inducers are beneficial in treating endometriosis…” was in mouse models and in-vitro studies unless there are references that can be added.
13. Line 534: “Te” looks like it should be “The.”
14. Lines 555-556: A reference is needed for “The regulation of ferroptosis can be utilized as an interventional target for treatment of endometriosis and disease-related infertility.”
15. Lines 561-562: The “optimal timing of treatment for ferroptosis modulators in humans” suggests that there is human data. Delete that or add references with human data.
Author Response
Answer to the reviewers
diagnostics-2314504
Title: Current understanding of and future directions for endometriosis-related infertility research with a focus on ferroptosis
Author: Hiroshi Kobayashi et al.
Dear Editor in Chief:
Thank you and the reviewers for the thoughtful comments and helpful suggestions on my manuscript “Current understanding of and future directions for endometriosis-related infertility research with a focus on ferroptosis” (manuscript ID: diagnostics-2314504), authored by Hiroshi Kobayashi et al. We have carefully considered each of the comments, made every effort to address the concerns raised, and applied corresponding revisions to the manuscript. Additionally, we have carefully revised the manuscript to ensure that the text is optimally phrased and free from typographical and grammatical errors. An English proofreading certificate by a native speaker was attached.
The detailed, point-by-point responses to the reviewer comments are given below, whereas the corresponding revisions are highlighted to our manuscript within the document.
We believe that our manuscript has been considerably improved as a result of this revision, and hope that the revised manuscript is acceptable for publication in Diagnostics.
We would like to thank you once again for your consideration of our work and inviting us to submit the revised manuscript. We look forward to hearing from you.
With best regards,
Hiroshi Kobayashi, M.D., Ph.D.
Department of Obstetrics and Gynecology, Nara Medical University
840 Shijo-cho, Kashihara, Nara, 634-8522, Japan
Tel: +81 744 29 8877
Fax: +81 744 23 6557
E-mail: hirokoba@naramed-u.ac.jp
An English proofreading certificate
Point-by-point responses to reviewer comments
Reviewer 1
Comment 1:
The authors present an overview of ferroptosis, an iron overload- and lipid-reactive oxygen species-dependent form of programmed cell death that is distinct from apoptosis, necrosis, and autophagy. The comments on treatment that suggest human data are overstated or under-referenced and need to be deleted or corrected.
Response 1:
The comments on treatment that suggest human data were inappropriate and the related texts were revised. See responses to comments 2, 10, 11, 12, 14 and 15.
Comment 2:
Lines 27-28: “indicating that the regulation of ferroptosis can be utilized as an interventional target for treatment…” suggests there is human data. Consider “suggesting that the regulation of ferroptosis should be studied as an interventional target for treatment…” If you have human data, please clarify that.
Response 2:
We fixed as you pointed out.
Endometriotic cells are characterized by ferroptosis resistance, whereas granulosa cells remain highly susceptible to ferroptosis, suggesting that the regulation of ferroptosis is utilized as an interventional target for the treatment of endometriosis and disease-related infertility.
Comment 3:
Line 38: “The presence of endometrial glands and stroma” suggests that endometriosis is endometrium. “Endometrial-like,” “endometriotic,” and “similar to endometrial” suggest that endometriosis is not endometrium but may come from the endometrium, Müllerian rests, tubal metaplasia, peritoneum, bone-marrow stem cells or other. Consider “endometrial-like,” “endometriotic,” or “similar to endometrial” glands and stroma.
Response 3:
We fixed as you pointed out.
Endometriosis is an estrogen-dependent disease characterized by the presence of endometrial-like glands and stroma outside the uterus [1].
Comment 4:
Line 42: “Endometriosis causes pelvic pain, dysmenorrhea, dyspareunia, and infertility.” Endometriosis does not cause all of those in all women with endometriosis. It can coincidental in fertile women at tubal ligation. Consider “can cause” in place of “causes.”
Response 4:
We fixed as you pointed out.
Endometriosis can cause pelvic pain, dysmenorrhea, dyspareunia, and infertility [1].
Comment 5:
Line 47-48: Reference 5 (Tanbo et al.)is not correct for “Retrograde menstruation and periodic hemorrhage from ectopic lesions is a hallmark feature of endometriosis, leading to adhesion and fibrosis of surrounding tissues.” Tanbo et al. have “Once established, these hormone-responsive and cyclically active endometriotic lesions drive acute then chronic inflammatory reactions, and lead to pelvic adhesions, pain, and infertility.” See comments on repeated tissue injury and repair (ReTIAR) below. Please correct the reference or the statement.
Response 5:
We changed to the following:
Tanbo and Fedorcsak reported that endometriosis can cause pelvic adhesions, pain, and infertility by inducing acute and chronic inflammation [3].
Comment 6:
Line 48, 493: Why don’t all women have adhesions if bleeding causes adhesions?
Response 6:
Platelet aggregation associated with hemorrhage is thought to be associated with abnormal adhesion, but the degree of platelet aggregation may vary among individuals. This may be the reason why all women do not have adhesions, but the details are unknown.
Comment 7:
Lines 225-228: Sun-Wei Guo would add that the blood may come from wounds undergoing repeated tissue injury and repair (ReTIAR).
Guo SW, Ding D, Shen M, Liu X. Dating endometriotic ovarian cysts based on the content of cyst fluid and its potential clinical implications. Reprod Sci. 2015 Jul;22(7):873-83. doi: 10.1177/1933719115570907. Epub 2015 Feb 11. PMID: 25676579; PMCID: PMC4565481.
Guo SW. Fibrogenesis resulting from cyclic bleeding: The Holy Grail of the natural history of ectopic endometrium. Hum Reprod. 2018, 33(3):353-356. doi: 10.1093/humrep/dey015. PMID: 29420711
Response 7:
There seems to be a mismatch between the question below and the number of lines pointed out by the reviewer. We added the following sentences to the first paragraph of the Introduction section:
Guo SW stated that endometriosis might be better redefined as “a condition that started with the ectopic deposition of endometrial stroma and epithelium which undergo cyclic bleeding and thus repeated tissue injury and repair, resulting in gradual and progressive smooth muscle metaplasia and fibrogenesis” [4].
Comment 8:
Lines 483-484: Change “infertility” to “mouse granulosa” in “Ni et al. demonstrated in infertility models…”
Response 8:
Ni et al. demonstrated in mouse granulosa models that iron chelators and vitamin E are useful therapies for the treatment of endometriosis-related infertility [27].
Comment 9:
Lines 512-513: Do you mean “ferroptosis as the pathophysiology of endometriosis” or “ferroptosis in the pathophysiology of endometriosis”?
Response 9:
Second, recent studies have focused on iron overload-dependent ferroptosis in the pathophysiology of endometriosis.
Comment 10:
Lines 513-514: Ref 13. Liang et al., is a mouse model. Change “The selective induction of ferroptosis has emerged as a new treatment strategy in endometriosis [13]” to reflect the mouse model.
Response 10:
We fixed it as you pointed out.
Selective induction of ferroptosis has emerged as a new treatment strategy in the mouse model of endometriosis [14].
Comment 11:
Line 519-520: There is no mention of treatment in reference 69, Ota et al. (2000). Reference 85, Wan et al. 2022, is not a treatment study. Delete or correct “Therefore, the activation of the ferroptosis pathways can be utilized as an interventional target for treatment of endometriosis.”
Response 11:
We deleted it as you pointed out.
Comment 12:
Line 523-524: “Overall, ferroptosis inducers are beneficial in treating endometriosis…” was in mouse models and in-vitro studies unless there are references that can be added.
Response 12:
We fixed it as you pointed out.
Overall, ferroptosis inducers are beneficial for the treatment of endometriosis in mouse models and in vitro studies.
Comment 13:
Line 534: “Te” looks like it should be “The.”
Response 13:
We fixed it as you pointed out.
Comment 14:
Lines 555-556: A reference is needed for “The regulation of ferroptosis can be utilized as an interventional target for treatment of endometriosis and disease-related infertility.”
Response 14:
We deleted this sentence.
Comment 15:
Lines 561-562: The “optimal timing of treatment for ferroptosis modulators in humans” suggests that there is human data. Delete that or add references with human data.
Response 15:
We fixed it as you pointed out.
Furthermore, the safety, tolerability, and efficacy of treatment with ferroptosis modulators in humans remain unknown.

Reviewer 2 Report
The manuscript of Hiroshi Kobayashi et al. “Current understandings and future directions for endometriosis-related infertility research focusing on ferroptosis” summarize current understanding of research in endometriosis and disease-related infertility, highlight the molecular mechanisms and signaling pathways involved in cell death termed ferroptosis, and finally discuss future therapeutic strategies. Overall, the manuscript is innovative and well written. However, there are few topics that fall short of expectation and needs further discussion. Here are my suggestions.
1 In this review, the “Materials and Methods” section in line 76 seems to be dispensable. And only 22 articles met the criteria after screening the articles using the authors' method, but 100 literatures were cited throughout. The authors do not mention or summarize these 22 articles that meet the criteria in detail in the later section, thus making this paragraph redundant.
2 The sub headings, “4.1. Iron overload in endometriosis” in line 224 and “4.2. Dysregulated antioxidant systems in endometriosis” in line 273, under heading “4. Oxidative-antioxidant balance as a biological markers of oocyte quality in endometriosis” in line 183 are not appropriate and not well explain the imbalance between oxidation and antioxidation. There seems to be a lack of passage about oxidation in endometriosis.
3The first sentence “This subsection summarizes the main cause of oxidative stress in endometriosis.” in line 225 shows that this passage is talking about oxidative stress, but the sub heading of this paragraph is “4.1. Iron overload in endometriosis”. The writer should clarify the concept of ferroptosis, iron overload and oxidative stress.
4 The passage “5. Ferroptosis” in line 324 appears too abrupt to be related to the previous text. The previous article has described endometriosis-related infertility and oxidative-antioxidant balance in endometriosis, and now suddenly turns to ferroptosis. There seems to be a lack of connection between these two parts.

Author Response
Answer to the reviewers
diagnostics-2314504
Title: Current understanding of and future directions for endometriosis-related infertility research with a focus on ferroptosis
Author: Hiroshi Kobayashi et al.
Dear Editor in Chief:
Thank you and the reviewers for the thoughtful comments and helpful suggestions on my manuscript “Current understanding of and future directions for endometriosis-related infertility research with a focus on ferroptosis” (manuscript ID: diagnostics-2314504), authored by Hiroshi Kobayashi et al. We have carefully considered each of the comments, made every effort to address the concerns raised, and applied corresponding revisions to the manuscript. Additionally, we have carefully revised the manuscript to ensure that the text is optimally phrased and free from typographical and grammatical errors. An English proofreading certificate by a native speaker was attached.
The detailed, point-by-point responses to the reviewer comments are given below, whereas the corresponding revisions are highlighted to our manuscript within the document.
We believe that our manuscript has been considerably improved as a result of this revision, and hope that the revised manuscript is acceptable for publication in Diagnostics.
We would like to thank you once again for your consideration of our work and inviting us to submit the revised manuscript. We look forward to hearing from you.
With best regards,
Hiroshi Kobayashi, M.D., Ph.D.
Department of Obstetrics and Gynecology, Nara Medical University
840 Shijo-cho, Kashihara, Nara, 634-8522, Japan
Tel: +81 744 29 8877
Fax: +81 744 23 6557
E-mail: hirokoba@naramed-u.ac.jp
An English proofreading certificate
Point-by-point responses to reviewer comments
Reviewer 2
Comment 1:
In this review, the “Materials and Methods” section in line 76 seems to be dispensable. And only 22 articles met the criteria after screening the articles using the authors' method, but 100 literatures were cited throughout. The authors do not mention or summarize these 22 articles that meet the criteria in detail in the later section, thus making this paragraph redundant.
Response 1:
We deleted "2. Materials and Methods", "2.1. Search Strategy and Selection Criteria", "Table 1" and "Figure 1".
Comment 2:
The sub headings, “4.1. Iron overload in endometriosis” in line 224 and “4.2. Dysregulated antioxidant systems in endometriosis” in line 273, under heading “4. Oxidative-antioxidant balance as a biological markers of oocyte quality in endometriosis” in line 183 are not appropriate and not well explain the imbalance between oxidation and antioxidation. There seems to be a lack of passage about oxidation in endometriosis.
Response 2:
The response to comment 2 will be answered together with comment 3.
See the end of Subsection 3.1.
Thus, follicular fluid oxidative stress may be useful as a diagnostic marker for oocyte quality prediction. However, inconsistency still exists, as Nakagawa et al. demonstrated that the oxidative stress status was not increased in the follicular fluid of patients with endometriosis [55].
Comment 3:
The first sentence “This subsection summarizes the main cause of oxidative stress in endometriosis.” in line 225 shows that this passage is talking about oxidative stress, but the sub heading of this paragraph is “4.1. Iron overload in endometriosis”. The writer should clarify the concept of ferroptosis, iron overload and oxidative stress.
Response 3:
To fully describe oxidative stress in endometriosis and clarify the concepts of ferroptosis, iron overload, and oxidative stress, the text was sorted in the following order:
- Oxidative-antioxidant (redox) balance in endometriosis
4.3. Biological markers involved in redox balance in endometriosis
4.1. Oxidative stress caused by iron overload
4.2. Dysregulated antioxidant systems
As you pointed out, we clarified the concept of ferroptosis, iron overload, and oxidative stress in the first paragraph of the section "5. Ferroptosis associated with endometriosis".
See the answer to the following question.
Comment 4:
The passage “5. Ferroptosis” in line 324 appears too abrupt to be related to the previous text. The previous article has described endometriosis-related infertility and oxidative-antioxidant balance in endometriosis, and now suddenly turns to ferroptosis. There seems to be a lack of connection between these two parts.
Response 4:
To explain the relationship between endometriosis and ferroptosis, the first paragraph of section 4 was modified as follows:
4.1. The ferroptosis-related pathway
Endometriosis is characterized by iron overload caused by ectopic periodic bleeding. Excess iron catalyzes the Fenton reaction to generate free radicals and then oxidative stress-induced LPO. Oxidative stress is the major cause of cell death, including apoptosis, autophagy, and ferroptosis, with the latter being an iron- and ROS-dependent form [16]. Therefore, endometriosis is thought to be closely associated with ferroptosis mediated by an iron-dependent oxidative stress [16,78,79]. Accumulating evidence suggests that ferroptosis is implicated in pathological conditions, such as cancer, neurodegenerative diseases, ischemic organ injuries, and pulmonary fibrosis [11,13,27,78,80]. With reference to these studies, we summarize the distinctive molecular mechanisms of sev-eral key pathways in the regulation of ferroptosis in endometriotic and granulosa cells as well as endometriosis-associated mac-rophages. Molecules that play key roles in ferroptosis are involved in metabolic and biochemical processes, including iron home-ostasis, LPO, glutathione metabolism, cysteine exploitation, mevalonate cholesterol biosynthesis, and nicotinamide adenine di-nucleotide phosphate (NADPH) function [81].

Round 2
Reviewer 1 Report
1. The authors are thanked and congratulated for extensive and excellent revisions to their paper and for clarifying that the data is not human. “Furthermore, the safety, tolerability, and efficacy of treatment with ferroptosis modulators in humans remain unknown” is correct.
2. The concluding lines of the abstract and the last word of the paper need to reflect those changes. Thus, changes are needed in three phrases in the last four sentences of the abstract and the last word of the paper. My suggestions follow. I am using the line numbers from a formatted redline version and do not have your original. I will send a note to the journal about this.
3. Line 28: “the regulation of ferroptosis is utilized as an interventional target for the treatment of endometriosis and disease-related infertility” is better as “the regulation of ferroptosis is utilized as an interventional target for research into the treatment of endometriosis and disease-related infertility.”
4. Line 30: “An analysis of the ferroptosis pathway enhances..” is better as “An analysis of the ferroptosis pathway in [(the lab) and/or (animal research) and/or (in vivo research) and/or (in vitro research)] enhances..” You will need to decide which of the bracketed modifiers is best. It needs to be clear that these are not human-based conclusions.
5. Line 31: “the role of ferroptosis modulators as novel treatments for endometriosis and disease-related infertility” is better as “the role of ferroptosis modulators as a research approach and potential novel treatment for endometriosis and disease-related infertility.”
6. Line 434 (the last word): "treatments" should be replaced by "research."
7. Line 428: This is good. Thank you. “Furthermore, the safety, tolerability, and efficacy of treatment with ferroptosis modulators in humans remain unknown.”
8. For future use, in 1984, I was criticized for not clarifying that my work was research and not clinically useful. Since that time, I have tried to make it clear when I was discussing research, even when I was using it clinically, and when I was discussing consensus clinical utility. If anything, I err on the side of calling it research. I would encourage you to call results research until it is clear they are clinical. Early adopters like me will accept it more readily and resistant late adopters will not be as defensive or aggressive; that is my experience. As a follow-up, the physician, a co-fellow in 1977, who was most critical in 1984, asked me to write a chapter in his textbook in 1986 and wrote the foreword in my 1990 textbook. We are still friends.
Author Response
Answer to the reviewers
diagnostics-2314504
Title: Current understanding of and future directions for endometriosis-related infertility research with a focus on ferroptosis
Author: Hiroshi Kobayashi et al.
Comments 3-6 have been corrected as indicated.
Regarding 8, the message from you will be my lesson from now on.
